# Grazing Horse Effects on Desert Grassland Soil Gross Nitrification and Denitrification Rates in Northern China

Xiaonan Wang [1], Chengjie Wang [2], Chengyang Zhou [1], Shining Zuo [1], Yixin Ji [1], Qiezhuo Lamao [1] and Ding Huang [1,*]

1    College of Grassland Science and Technology, China Agricultural University, Beijing 100193, China; b20203240974@cau.edu.cn (X.W.); s20193040615@cau.edu.cn (C.Z.); s20203243178@cau.edu.cn (S.Z.); s20213243257@cau.edu.cn (Y.J.); sy20213243290@cau.edu.cn (Q.L.)

2    College of Grassland, Resources and Environment, Inner Mongolia Agricultural University, Hohhot 010011, China; nmgcjwang3@imau.edu.cn

*    Correspondence: huangding@cau.edu.cn

**Abstract:** The aim of this study was to investigate the effects of grazing on soil gross nitrification (GN) and denitrification (DN) rates and soil environmental factors on GN and DN in the desert grassland of northern China. Soil samples were collected from July to November in 2017 and March to June in 2018, with 5-5 soil samples taken from three enclosures (CK) vs. three heavy-grazing (G) randomized treatment blocks. We determined: (1) the soil moisture (SM), pH, bulk density (BD), total nitrogen (TN), soil organic carbon (SOC), and inorganic nitrogen (IN, $NH_4^+$-N, and $NO_3^-$-N) content, and (2) GN and DN. The relationship between the changes in GN, DN, and the soil environment was analyzed using stepwise multiple-regression analysis. Gross nitrification, DN, pH, BD, C/N, SM, IN, and $NO_3^-$-N varied significantly by month. Grazing induced significant increases in SM and $NO_3^-$ only. GN in the CK treatment was related to $NH_4^+$-N and $NO_3^-$-N, while GN in the G treatment was related to $NH_4^+$-N and SM. DN in the CK treatment was related to $NH_4^+$-N, while DN in the G treatment was related to C/N. Additionally, GN and DN had obvious seasonal variations and reached a maximum in July. This highlights the different underlying mechanisms that affect soil GN and DN and the dynamics, particularly in the desert grassland system.

**Keywords:** gross nitrification; denitrification; grazing; desert grassland; seasonal variations

## 1. Introduction

Nitrogen (N) plays an important role in plant growth [1,2]. Gross nitrification (GN) and denitrification (DN), as two vital processes of soil N cycling, regulate the form of soil inorganic nitrogen (IN, $NH_4^+$ and $NO_3^-$) available to plants [3,4]. GN is the transition from ammonium nitrogen ($NH_4^+$) to nitrate nitrogen ($NO_3^-$), while DN is the transition from $NO_3^-$ to nitrogen gas $N_2O$ and $N_2$ [5]. Grasslands cover 26% of global terrestrial surface [6], and grazing is one of the main ways of using grasslands. Animal grazing increases soil compaction through trampling [7] and affects grassland nutrient cycling through feed intake and excreta return [8]. These effects have been shown to influence GN and DN.

Increasingly, studies have been conducted in grasslands assessing GN under different grazing managements [4,9–12]. While some studies have found that grazing promotes GN [13,14], others have found that grazing inhibits soil GN [15,16]. Numerous studies have been conducted on grassland soil DN under different grazing management conditions [17–20]. While Du R et al. and Gao [21,22] found that grazing promotes soil DN, Sun et al. and Wang et al. concluded that grazing inhibits soil DN [11,23]. From previous research conducted, no consistent relationships were found between animal grazing effects and soil GN and DN.

Soil GN can be influenced by soil physicochemical properties, such as soil carbon:nitrogen (C:N) ratios and soil moisture [24–26]. For example, there is a negative correlation between GN and soil C/N, because micro-organisms change N by using C [27]. Li et al. found that soil organic carbon (SOC) decreases the GN rate [28], whereas Luo et al. found that the GN rate increases as the SOC content increases [29]. As the substrate of GN, the content of $NH_4^+$ would directly affect the rate of GN. Generally, the higher the $NH_4^+$ content is, the faster the GN rate is [30]; however, another study found that the GN rate does not increase with the increase in $NH_4^+$ content [31]. In the upper 100 cm of soil, soil total N is mainly composed of organic N [32]. Organic N provides a substrate for heterotrophic nitrification [33]. Yao et al. found that the higher the organic N content in soil, the greater the GN [34]. Soil pH affects GN mainly by affecting the composition and function of micro-organisms [35,36]. When soil pH > 7.5, the GN rate of adding anhydrous ammonia reached 89%, whereas when the pH < 6, it was 39% [37]. In addition, the rate of GN was increased by increasing the soil pH by adding lime [38]. The soil GN rate was affected by temperature, but there was no consistent relationship. Gao et al. found that the GN rate decreased with the increase in temperature [39], whereas Bork et al. found that the increase in temperature greatly promoted the GN rate [40]. Soil GN is regulated by precipitation through changes in nitrate [41]. Chen et al. found that the higher the soil water content, the faster the rate of GN [42]. However, it was found that the GN rate decreased with the increase in soil moisture content [43].

Soil DN occurs under anaerobic conditions and is affected by soil environmental factors [44–46]. For example, $NO_3^-$, as the substrate of DN, and SOC, as the electron donor of denitrifying bacteria, affect the DN rate [47,48]. The addition of $NO_3^-$ and C stimulates the DN rate [49]. The effect of soil pH on DN was not consistent. In general, acidic soil inhibits DN [50], while alkaline soil promotes DN [51]. In addition, there was no significant relationship between pH and DN [52]. DN is also affected by temperature. In general, the DN rate increases as the temperature increases [53]. Conversely, the DN rate would decrease if the temperature was higher or lower than the optimal temperature [54].

Here, we investigated the effects of animal grazing on soil GN and DN from the desert grassland of Inner Mongolia, northern China. Previous studies have covered the impact of animal grazing on plant productivity and community characteristics [55,56] and soil physical and chemical properties [57,58]. However, no information is available on soil GN and DN in desert grasslands in China. The aim of this study was to determine the impact of animal grazing on soil GN and DN over a year (except winter) in desert grassland soil. Specifically, we aimed to determine if animal grazing influences soil GN and DN.

## 2. Materials and Methods

### 2.1. Study Sites

The experiment was conducted in the Xilamuren desert steppe (41°21′ N, 111°112′ E), near Damao County of Baotou City, Inner Mongolia, China. The climate in this region is mid-temperate semi-arid continental monsoon. The elevation is about 1600 m. The annual precipitation is 280 mm with abundant precipitation in summer, from July to September. The annual evaporation is about 2300 mm. The annual average temperature is 2.5 °C. The soil type is loamy sand texture (using the FAO classification). The experimental grassland was dominated by *Stipa krylovii*. Other common species include *Stipa breviflora*, *Cleistogenes songorica*, and *Artemisia frigida*.

### 2.2. Experimental Design

In 2012, an 8.64 ha experimental site was established with relatively homogenous vegetation, which was previously free-grazing land, and the grassland area around the experimental site was about 70,000 ha. The experimental area was divided into two treatments (CK: enclosure, and G: heavy grazing with a stocking rate of 1.05 sheep/ha/year (5 horses)) with three replicates per treatment in a randomized complete block design. For

the grazing treatments, adult Mongolian horses with similar body weight and age grazed for the first 5 days of each month, from June to September each year, from 2012 to 2018.

### 2.3. Soil Sampling

Soil samples were collected from July to November in 2017 and March to June in 2018. Five intact soil cores were randomly collected with a soil corer at a depth of 10–15 cm in each plot (four points at the corners and one point in the center). Soil samples kept in an incubator with ice bags were brought back to the laboratory and immediately transferred to a 4 °C refrigerator for storage. Soil GN and DN rates were determined within 7 days of sampling. In addition, we collected additional soil samples to determine soil pH, bulk density (BD), soil moisture (SM), total nitrogen (TN), $NH_4^+$-N, $NO_3^-$-N, and IN and SOC. Soil temperature (ST) was automatically monitored by the PC-4 Meteorological environment monitoring recorder (PC-4, Jinzhou Sunshine Meteorological Technology Co., Ltd., Jinzhou, China).

### 2.4. Soil Characteristic Measurements

The GN and DN rates of the soil were determined using Barometric Process Separation Technology (BaPS, Aozuo Ecological Instrument Co., Ltd., Beijing, China). Five intact soil cores from each plot were placed in the BaPS incubation chamber to determine the rates of GN and DN. Following Conrads et al. to overcome the measurement error of BaPS for alkaline calcareous soil, the respiration quotient was set to 0.9 [59]. The data collection time was over 12 h, and the system automatically recalculated the soil GN rate and DN rate. Soil pH was measured using a pH meter (PHS-3C, Shanghai INESA Scientific Instrument Co., Ltd., Shanghai, China) with a soil–water mass ratio of 1:2.5. BD was measured using the soil core method at the subsurface depth (10–15 cm) using the BaPS soil core sampler, and SM was measured using the gravimetric method. $NH_4^+$-N and $NO_3^-$-N were determined by a continuous flow analyzer (AA3, Tianjin Zhongtong Technology Development Co., Ltd., Tianjin, China) after extracting fresh soil with 2 mol/L KCl. SOC and TN were determined by an elemental analyzer after dilution with hydrochloric acid to remove calcium carbonate (Vario MACRO cube, Beijing Zhongke Huaxing Technology trade Co., Ltd., Beijing, China).

### 2.5. Statistical Analysis

To test the effects of treatment, month, and their interactions on GN, DN, and soil environmental factors, we used a repeated-measures mixed-model analysis of variance with treatment as the fixed effect, and month as a repeated-measures factor. To compare and analyze the differences between CK and G treatments, the response ratio (RR) was used to reflect the response effect of each index to grazing. The specific calculation formula [60,61] is as follows:

$$RR = ln\left(\frac{Xt}{Xc}\right) = ln\ Xt - ln\ Xc \tag{1}$$

where *Xt* represents the average value of the G treatment and *Xc* represents the average value of the CK treatment. When *RR* was 0, it indicated that grazing did not cause differences in parameters between the G and CK groups. If *RR* was less than 0, it meant that the grazing activity had a negative effect on the selected parameter; if *RR* was greater than 0, it meant that the grazing activity had a positive effect on the selected parameter. The differences in soil GN and DN between the CK and G treatments were tested using a *t*-test. Pearson's correlation analysis and linear regression analysis were used to test relationships between the soil environmental factors and GN and DN under different treatments. Stepwise multiple-regression analyses were performed with soil physicochemical properties as independent variables and GN and DN as dependent variables. All statistical analyses were conducted with SPSS 19.0 statistical software package (SPSS, Chicago, IL, USA), and figures were plotted with SigmaPlot 12.5 (Systat Software, Inc., San Jose, CA, USA).

## 3. Results

### 3.1. Soil Analysis

SM and $NO_3^-$-N content showed significant differences between the CK and G treatments ($p < 0.05$), but no significant difference for the other indices ($p > 0.05$; Table 1). In contrast, GN, pH, BD, C/N, SM, IN, and $NO_3^-$-N showed strong significant differences between months ($p < 0.001$) and DN showed a slight significant difference between months ($p < 0.05$; Table 1). There were interactions between treatments and months for pH, $NH_4^+$-N, and BD ($p < 0.05$ or $p < 0.01$; Table 1). Grazing response on SOC, $NO_3^-$-N, IN, SM, C/N, BD, and TN showed an increasing trend, but GN, DN, and $NH_4^+$-N had a decreasing trend (Figure 1).

**Table 1.** Effects of treatment (T), month (M), and their interaction on gross nitrification (GN), denitrification (DN), and soil physical and chemical properties in a desert steppe, based on a repeated-measures analysis of variance.

| | Treatment | | Month | | T * M | |
|---|---|---|---|---|---|---|
| | **F** | **P** | **F** | **P** | **F** | **P** |
| GN | 5.67 | 0.14 | 7.198 | *** | 0.736 | 0.66 |
| DN | 2.316 | 0.267 | 3.251 | * | 0.273 | 0.966 |
| pH | 2.888 | 0.231 | 11.156 | *** | 3.558 | * |
| $NH_4^+$-N | 12.832 | 0.07 | 2.319 | 0.072 | 4.594 | ** |
| TN | 17.379 | 0.053 | 1.218 | 0.349 | 0.646 | 0.729 |
| BD | 5.589 | 0.142 | 8.839 | *** | 2.644 | * |
| C/N | 17.729 | 0.052 | 18.633 | *** | 0.594 | 0.769 |
| SM | 26.094 | * | 87.895 | *** | 1.947 | 0.122 |
| IN | 18.6 | 0.05 | 10.799 | *** | 1.645 | 0.189 |
| $NO_3^-$-N | 21 | * | 11.567 | *** | 2.272 | 0.077 |
| SOC | 16.807 | 0.055 | 1.904 | 0.13 | 0.315 | 0.949 |

Unmarked letters, *, **, *** represent $p < 0.05$, $p < 0.01$, and $p < 0.001$ respectively. GN: gross nitrification; DN: denitrification; TN: total nitrogen; BD: bulk density; C/N: SOC/TN; SM: soil moisture; IN: inorganic nitrogen; SOC: soil organic carbon.

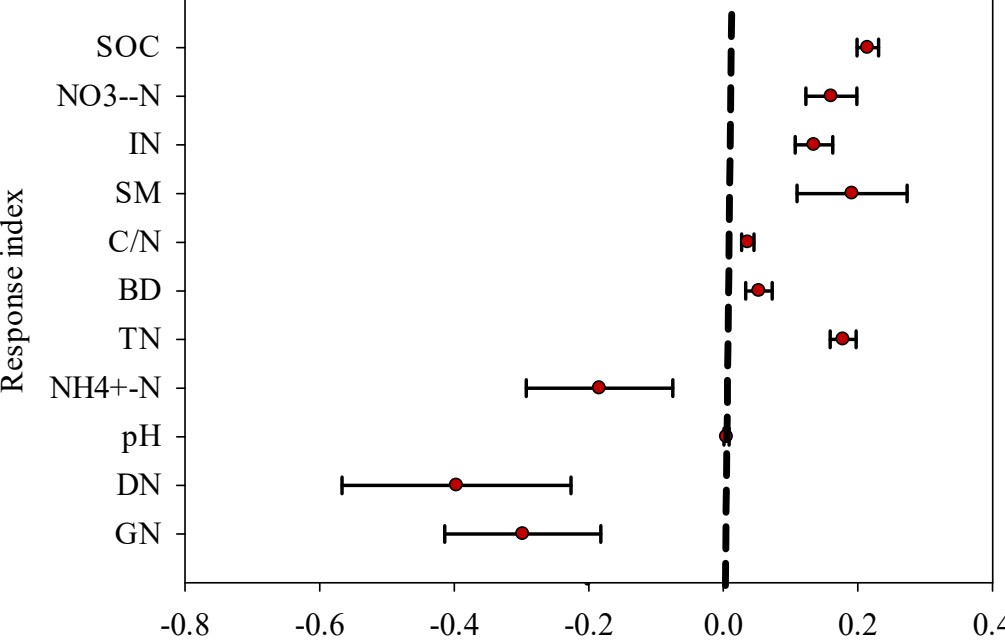

**Figure 1.** Gross nitrification (GN), denitrification (DN), and soil physical and chemical properties as responses to grazing.

### 3.2. Soil GN and DN

GN was significantly lower ($p < 0.05$) in the G than CK treatment in July (140.04 and 173.17 ug kg$^{-1}$ h$^{-1}$, respectively, Figure 2a), and varied significantly between months ($p < 0.001$), but no interaction was observed between grazing treatment and months. DN was significantly lower ($p < 0.05$) in the G than CK treatment in July (107.00 and 135.98 ug kg$^{-1}$ h$^{-1}$, respectively, Figure 2b), and varied significantly between months ($p < 0.05$), but no interaction was observed between grazing treatment and months.

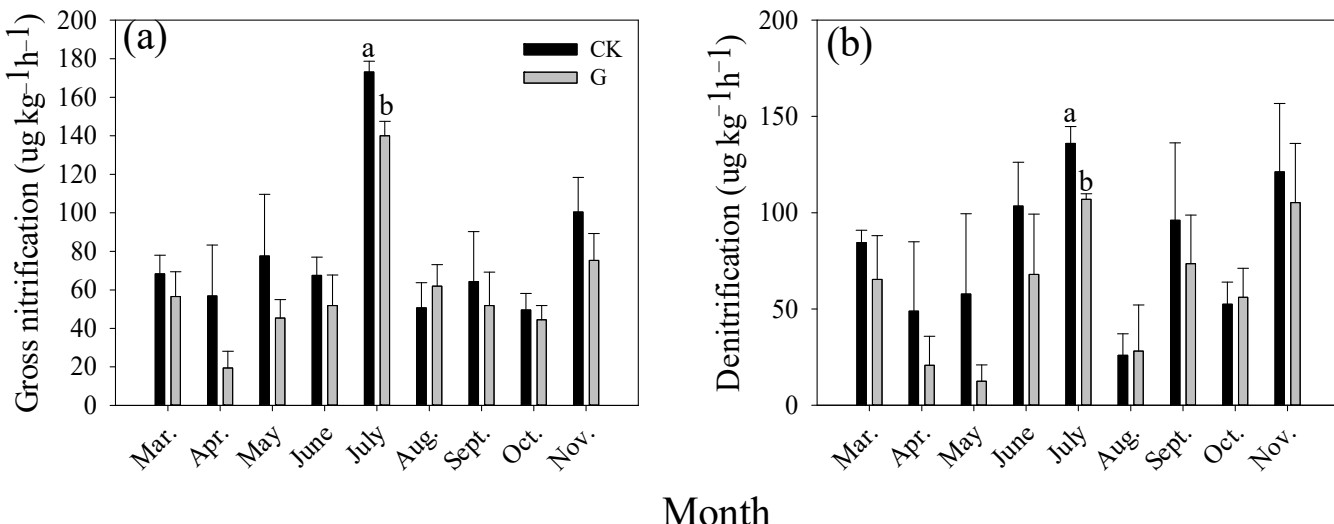

**Figure 2.** Monthly gross nitrification (GN) (**a**) and denitrification (DN) (**b**) in enclosure (CK) and heavy grazing (G) treatments. Different lowercase letters indicate a significant effect between treatments ($p < 0.05$).

### 3.3. Correlations between GN, DN, and Soil Physical and Chemical Properties

GN in the CK treatment showed a significant positive correlation with $NH_4^+$-N ($R^2 = 0.214$; $p < 0.05$) and with SM ($R^2 = 0.182$; $p < 0.05$) but showed a significant negative correlation with $NO_3^-$-N ($R^2 = 0.274$; $p < 0.01$, Figure 3). DN in the CK treatment showed a significant positive correlation with $NH_4^+$-N ($R^2 = 0.198$; $p < 0.05$). GN in the G treatment showed a significant negative correlation with $NH_4^+$-N ($R^2 = 0.248$; $p < 0.01$) and C/N ($R^2 = 0.156$; $p < 0.05$), but showed a significant positive correlation with SM ($R^2 = 0.303$; $p < 0.01$). DN in the G treatment showed a significant negative correlation with C/N ($R^2 = 0.17$; $p < 0.05$).

### 3.4. Factors Controlling GN and DN Processes

GN in the CK treatment was positively correlated with $NH_4^+$-N, but was negatively correlated with $NO_3^-$-N; $NH_4^+$-N and $NO_3^-$-N as explanatory variables could explain 41.5% of the variance in the dependent variable GN in the CK ($p < 0.01$; Table 2). DN in the CK treatment was positively correlated with $NH_4^+$-N and $NH_4^+$-N could explain 19.8% of the variance in the dependent variable DN in the CK ($p < 0.05$; Table 2). GN in the G treatment was positively correlated with SM, but was negatively correlated with $NH_4^+$-N; SM and $NH_4^+$-N as explanatory variables could explain 53.3% of the variance in the dependent variable GN in the G ($p < 0.001$; Table 2). DN in the G treatment was negatively correlated with C/N, and C/N could explain 17% of the variance in the dependent variable DN in the G ($p < 0.05$; Table 2).

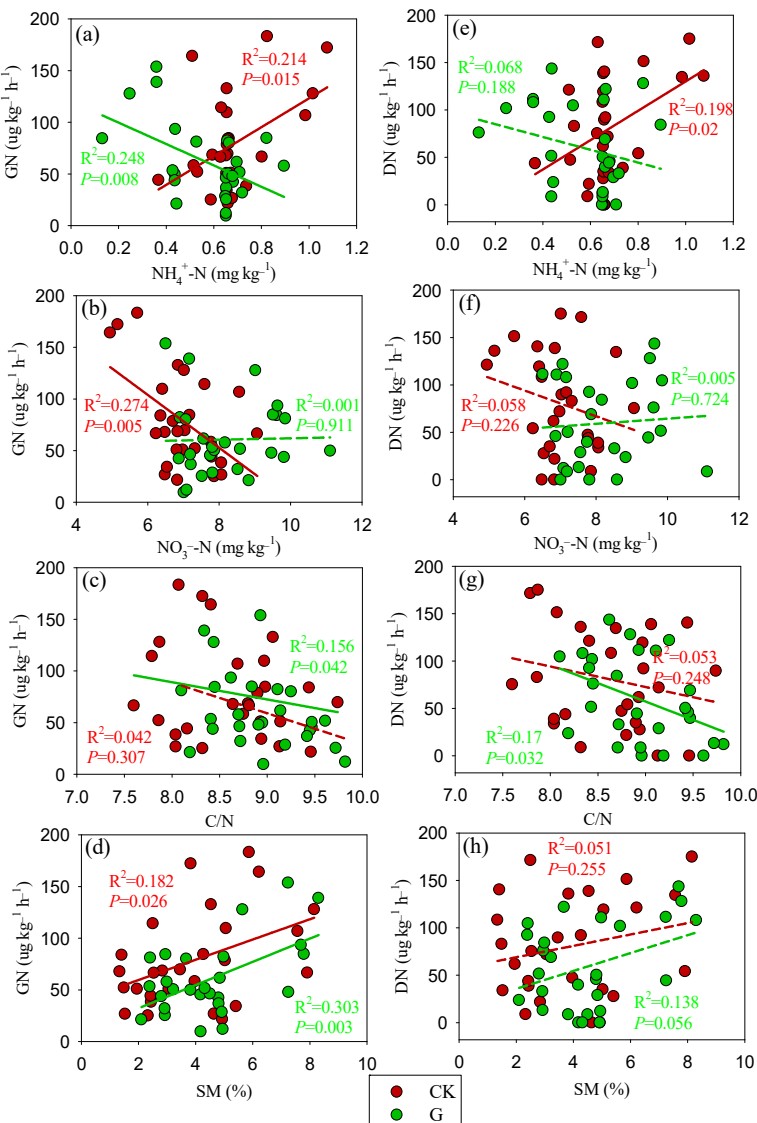

**Figure 3.** Relationships between gross nitrification (GN) and $NH_4^+$-N (**a**), $NO_3^-$-N (**b**), SOC/TN (C/N) (**c**), and soil moisture (SM) (**d**) in enclosure (CK) and heavy grazing (G) treatments, respectively. Relationships between denitrification (DN) and $NH_4^+$-N (**e**), $NO_3^-$-N (**f**), SOC/TN (C/N) (**g**), and soil moisture (SM) (**h**) in enclosure (CK) and heavy grazing (G) treatments, respectively. $R^2$ is the coefficient of determination.

**Table 2.** Stepwise multiple regression analysis of the soil gross nitrification (GN) and denitrification (DN) rates against the measured soil parameters across the study sites.

| Treatment | Dependent Variables | Standard Coefficient Regression Equation | *F* Value | $R^2$ | *p* Value |
|---|---|---|---|---|---|
| CK | GN | $Y = -0.455NO_3^-$-$N + 0.382NH_4^+$-$N$ | 8.51 | 0.415 | 0.002 |
| | DN | $Y = 0.445NH_4^+$-$N$ | 6.170 | 0.198 | 0.02 |
| G | GN | $Y = 0.535SM - 0.480NH_4^+$-$N$ | 13.701 | 0.533 | 0 |
| | DN | $Y = -0.413C/N$ | 5.130 | 0.17 | 0.032 |

CK: enclosure; G: heavy grazing; GN: gross nitrification; DN: denitrification; SM: soil moisture; C/N: soil organic carbon/total nitrogen.

## 4. Discussion

### 4.1. Grazing Effects on GN and DN in Desert-Steppe Grassland

Our results showed that GN in the CK treatment was positively correlated with $NH_4^+$-N and SM (Figure 3; Table 2), but was negatively correlated with $NO_3^-$-N (Figure 3;

Table 2). Studies found that SM was the main factor affecting soil GN [2,62]. In this study, we observed positive effects between GN and SM. It was found that the higher the SM, the faster the GN rate [26,42], whereas Vernimmen et al. found that with the increase in SM, the GN rate decreased [43]. Studies have shown that there is critical water content in the GN rate. Below the critical water content, the soil GN rate increases with an increase in soil water content. Above the critical water content, the GN rate decreases with an increase in water content [63,64]. This study area belongs to the driest part of the steppe, with an average annual precipitation of about 280 mm, mainly in the form of rainfall in summer. Therefore, for the majority of the year, the soil is under water deficit. $NH_4^+$-N is used as a substrate for GN, and its concentration in soil directly affects the intensity of GN [30,65]. We observed strong positive effects between GN and $NH_4^+$-N. It was found that the GN rate increased with the increase in $NH_4^+$-N [66,67]. However, this is in contrast with other studies that found GN was not significantly correlated with $NH_4^+$-N [2,31,68]. $NO_3^-$-N is a product of soil GN, and we found strong negative effects between GN and $NO_3^-$-N. A high concentration of product ($NO_3^-$-N) mainly inhibits the growth of nitrite bacteria [69], thus hindering the nitrification process. GN in the G treatment was positively correlated with SM (Figure 3; Table 2), but was negatively correlated with $NH_4^+$-N, C/N, and pH (Figure 3; Table 2). GN rates increased significantly with the increase in SM in the G treatment ($R^2$ = 0.303, $p$ = 0.003), although the SM after grazing was higher than that in the enclosures (Figure 1), and the SM after grazing was within the lower soil water content range. Soil C:N has an important effect on GN, affecting the abundance of ammonia oxidizing bacteria and archaea [70,71]. In this study, GN and C/N showed a negative correlation (Figure 3). This is consistent with previous research results showing that forest soil C:N is negatively correlated with GN [70,72]. The higher C/N ratio of grassland soil would increase the demand of micro-organisms for N, leading to a reduction in available substrates for GN, thus reducing the GN rate [73]. Soil pH affects the GN rate mainly by affecting the activity of soil-nitrifying bacteria and the GN process [74]. In our study, the pH was 7.57–7.99, with negative effects between GN and pH in the G treatment (Table S2). Conversely, when soil pH is lower than 4.5 or higher than 8.0, GN is inhibited [75]. When the soil pH is between 7.0 and 8.0, the GN process is promoted [76,77]. This discrepancy may be attributed to differences in grazing livestock, stocking rates, and grassland types. In addition, we found strong negative effects between GN and $NH_4^+$-N in the G treatment (Figure 3 and Table 2). We speculate that the main reason for these adverse results between enclosure and grazing treatment is the change in microbial activity under grazing pressure [78,79].

$NH_4^+$-N has a certain influence on DN. Our results showed that DN in the CK treatment was positively correlated with $NH_4^+$-N (Figure 3; Table 2). Mulvaney et al. showed that $NH_4^+$-N fertilizer can enhance DN by affecting the content of water-soluble organic C and pH in the soil [80]. In contrast, Wang et al. found that adding a small amount of $NH_4^+$-N inhibited the DN rate [81]. This discrepancy may also be attributed to difference in microbial activity.

In addition, DN rates are regulated by soil C and N content [82,83]. Our results showed that DN in the G treatment was negatively correlated with C/N and SOC (Figure 3; Table 2; Table S2). This may be because increasing the water content could increase the SOC and TN content in the G treatment [84,85]. In addition, other experiments in the desert steppe of Inner Mongolia have shown that short-term heavy grazing increased microbial nitrogen retention [86], which in turn reduced DN [87]. Yan et al. showed that heavy grazing can reduce C and N sequestration potential [88]. This discrepancy in findings may be due to differences in climate, soil physicochemical properties, and microbial composition in the study area itself, or due to changes in plant and microbial communities caused by grazing, resulting in different degrees of response.

### 4.2. Seasonal Dynamics of GN and DN

The monthly soil GN and DN rates showed strong seasonal variations. This agrees with previous studies on the significant seasonal variation of soil GN and DN rates in

different geographic regions [14,89]. Soil GN and DN rates are affected by a variety of factors, among which seasonal changes in SM, temperature, and IN can affect N conversion [2,67,90]. During the sampling period, the maximum soil GN and DN rates occurred in July, and soil GN was related to SM, $NH_4^+$-N, and $NO_3^-$-N. During this period, SM was the second highest in July after November (Figure S1), coupled with suitable $NH_4^+$-N, $NO_3^-$-N, temperature, and better hydrothermal conditions, which made the microbial activity and growth stronger and promoted GN [91]. Soil GN rates were significantly positively correlated with SM, but soil temperature had no significant effect on the soil GN rate (Tables S1 and S2). Therefore, during the sampling period, temperature was not the main influencing factor restricting soil GN in the grassland. The experimental area is in a semi-arid climate zone, and the SM remains low for extended periods. During our experiments, SM content was below 10% throughout the sampling period, while the soil temperature fluctuated between −1.53 and 24.33 °C (Figure S1). Therefore, although soil temperature conditions in July (24.33 °C) were suitable for soil nitrogen transformation, due to the limiting effect of lower SM, the effect of soil temperature was reduced, the effect of SM on soil GN was significant, and it became a main limiting factor affecting the change of soil GN rate. The soil GN and DN rates in November were the second highest. It may be that the soil underwent freezing and thawing, and the soil water content had reached its highest level (Figure S1). In addition, the content of $NH_4^+$-N and $NO_3^-$-N was also high, which promoted soil GN, but this may also be due to the soil temperature dropping to below zero (−1.53 °C). Therefore, the GN in November was lower than that in July. Since soil DN was positively correlated with GN (Tables S1 and S2), soil DN showed a consistent seasonal trend, reaching a maximum in July, followed by November.

## 5. Conclusions

Despite having no effect on GN and DN, there were different pathways affecting GN and DN under grazing pressure compared with CK. The soil GN rate in the CK treatment was regulated by $NH_4^+$-N and $NO_3^-$-N, while the soil GN rate in the G treatment was affected by $NH_4^+$-N and SM. Likewise, the DN rate in the CK treatment was regulated by $NH_4^+$-N, while the DN rate in the G treatment was regulated by C/N. Additionally, the soil GN and DN rates in the desert steppe of Inner Mongolia had obvious seasonal variations and reached a maximum in July. This highlights the different underlying mechanisms which affect soil GN and DN and their dynamics, particularly in the desert grassland system. Additionally, since GN and DN in soil are processes involving micro-organisms, it is necessary to study the micro-organisms related to GN and DN. Understanding their underlying effects may be critical to clarify the response mechanism of GN and DN to grazing and lay a foundation for in-depth understanding of soil N cycle in a desert steppe ecosystem.

**Supplementary Materials:** The following supporting information can be downloaded at: https://www.mdpi.com/article/10.3390/agriculture12071036/s1, Figure S1: Monthly changes of surface soil temperature and moisture in the experimental site; Table S1: Correlation analysis between gross nitrification (GN), denitrification (DN) and soil physical and chemical properties in the CK treatment; Table S2: Correlation analysis between gross nitrification (GN), denitrification (DN) and soil physical and chemical properties in the G treatment.

**Author Contributions:** Investigation, X.W., C.Z., S.Z., Y.J. and Q.L.; project administration, C.W., D.H. and X.W.; resources, C.W.; visualization, X.W.; writing—original draft, X.W.; writing—review and editing, D.H.; funding acquisition, D.H. All authors have read and agreed to the published version of the manuscript.

**Funding:** This research was funded by the Beijing Scientific Committee project of China, grant number Z181100009618031; the Climate-Smart Grassland Ecosystem Management Project, grant number 10006-P166853-2021-PIR-WB-China.

**Institutional Review Board Statement:** Not applicable.

**Informed Consent Statement:** Not applicable.

**Data Availability Statement:** The data presented in this study are available on request from the corresponding author.

**Conflicts of Interest:** The authors declare no conflict of interest.

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
