# Peer review of "Grazing Horse Effects on Desert Grassland Soil Gross Nitrification and Denitrification Rates in Northern China"

_agriculture, doi:10.3390/agriculture12071036_

Round 1

Reviewer 1 Report

I have attached my comments.

Reviewer 2 Report

This is an interesting and valuable study providing important and novel information about two vital processes of soil N cycling. The paper is written in good English with clear structure, good logic and good discussion. Sampling design and analyses seems to be appropriate. However, adding information about land use history and landscape context and clarifying the effects of data structure on the correlation and regression results would be necessary and would increase the impact of this contribution. I missed the discussion why soil moisture increased significantly due to heavy grazing (I would expect the opposite trend). I also suggest improving the explanation about the negative correlation between soil NH4+ and GN in G treatment. NH4+ is used as a substrate for GN that implies positive correlation. This expected positive correlation was really found in CK treatment. However, it converted to negative relationship after short-term heavy horse grazing. Authors suggest that livestock urine increases soil NH4+ and thereof soil pH. Increased soil pH then promotes NH3 formation that is toxic for nitrifying bacteria (cf. present discussion on L 228-233). This could be a logical explanation in general. However, in the present study pH did not changed due to horse grazing (cf. Table 1, Fig 1). Therefore, the pH based explanation cannot be appropriate here. Please, revise and give alternative explanation. Also consider that that magnitude of soil NH4+ variability was similar in CK and G treatments (cf. Fig.3a) and soil NH4+ did not increased due to grazing (cf. Table 1). I also suggest revising the term used for relationships (interactions) in the whole manuscript. The terms “regulation”, “control”, “effect”, “correlation“, “response” are not synonyms. For example, regulation assume negative feed backs.

Detailed comments

Title: The type of grazing animal might be important for N dynamics. Therefore, I suggest noting the grazing animal (horse) in title. In fact not many papers analyzed the effects of horse grazing so such new title could be more attractive.

Abstract: I suggest clarifying the information of this sentence: “Soil samples were collected during 2017 and 2018, with 3 pairs of soil samples taken from enclosure (CK) vs. heavy grazing (G) treatments for each sample.” in L 15-16. It now suggests two replicated years of seasonal sampling.

Suggested version: “Soil samples were collected from July to November in 2017 and March to June in 2018, with 5-5 soil samples taken from 3 enclosures (CK) vs. 3 heavy grazing (G) randomized treatments blocks.”

Introduction: L76 “However, no information is available on soil GN and DN.” This statement is not clear. Please, consider the literature you cited and referred (cf. references: [4] to [23]). I suggest specifying this statement on novelty: “However, no information is available on soil GN and DN in desert grasslands in China.”

Experimental design: It would be important to know details of degradation status and soil characteristics and vegetation characteristics at the start of the experiment. How big is the grassland site around the experimental area? Is the stocking rate defined for ships equal with the specific type of horse grazing? Please, clarify. Other studies used to define stocking rate by the number of livestock units (referring to cattle as livestock). Any case, different animals graze differently and emphasizing specificity of horse grazing would be important here.

Soil sampling: Please, add information how soil temperature was measured.

Statistical analysis: L 121 “To test the effects of treatment, month, and their interactions with GN, DN….” is misleading. I suggest: “To test the effects of treatment, month, and their interactions on GN, DN….”

Please, clarify how “average” was calculated to response ratio (average of spatial replicates in separated dates or average of temporal replicates at each sample points)? What is the interpretation of variation (size of horizontal line) on Figure 1?

You considered the dependence of temporal replicates using repeated measure mixed model. However, other correlation and regression analyses assume independence of samples. Please, add details how the spatial and temporal autocorrelations (dependences) were considered in these analyses?

Results:

L 154-155 “different lowercase letters indicate….” This is not relevant in Table 1 but this note is missing from Figure 2. Please, correct.

It is not clear how results of Table 1 and Figure 1 are related. For example, response ratio of SOC is higher than response ratio of NO3- (Figure 1), still SOC response was not significant while NO3- response was significant to grazing (cf Table 1). Please, clarify.

Caption to Figure 1. “properties of grazing” should be “properties as responses to grazing”.

Figure 3. Units are missing from all subfigures. Please, correct.

L185 when reporting results of stepwise multiple regression you should refer “dependent variables” (GN, DN) and “driving or explanatory variables” (NO3-, NH4+, SM etc…), i.e. referring just “correlation” is not precise here.

Discussion: In general, it is excellent and interesting. However, some argumentations were not convincing and need additional clarification.

L Negative correlation between GN and NO3-. Please, consider to discuss if this correlation might reflect really a regulatory process when the increasing concentration of the product of nitrification (NO3-) might have really a negative feedback and decreases the rate of GN.

L 216 Please, explain why SM was larger in the grazing treatment comparing to CK.

L 226-228 please, revise. If pH between 7 and 8 has positive effect (L 226-227) it cannot explain the negative effect you detected within this pH range in this study (L 228).

Similarly, the explanation in L 228-233 is not convincing (see my comments above). Please, revise.

L 238-240. This explanation is not convincing. If NO3- is a substrate for denitrification, please, explain why NO3- is reducing DN? In fact, the positive correlation between NH4+ and DN appears only in CK. Please, explain why this relationship changes and become non significant in G ( at similar magnitude of NH4+ variation).

Conclusions: It is OK. However, I miss the summary and explicit statements about grazing effect (that was the main focus of this study). Please, add highlights reporting about results in Figure 1, Figure 2 and Table 1. I missed the statement that GN and DN decreased due to grazing.
